

# False memories when viewing overlapping scenes

Filip Děchtěrenko and Jiří Lukavský

Institute of Psychology, Czech Academy of Sciences, Prague, Czech Republic

## ABSTRACT

Humans can memorize and later recognize many objects and complex scenes. In this study, we prepared large photographs and presented participants with only partial views to test the fidelity of their memories. The unpresented parts of the photographs were used as a source of distractors with similar semantic and perceptual information. Additionally, we presented overlapping views to determine whether the second presentation provided a memory advantage for later recognition tests. Experiment 1 ($N = 28$) showed that while people were good at recognizing presented content and identifying new foils, they showed a remarkable level of uncertainty about foils selected from the unseen parts of presented photographs (false alarm, 59%). The recognition accuracy was higher for the parts that were shown twice, irrespective of whether the same identical photograph was viewed twice or whether two photographs with overlapping content were observed. In Experiment 2 ($N = 28$), the memorability of the large image was estimated by a pre-trained deep neural network. Neither the recognition accuracy for an image part nor the tendency for false alarms correlated with the memorability. Finally, in Experiment 3 ($N = 21$), we repeated the experiment while measuring eye movements. Fixations were biased toward the center of the original large photograph in the first presentation, and this bias was repeated during the second presentation in both identical and overlapping views. Altogether, our experiments show that people recognize parts of remembered photographs, but they find it difficult to reject foils from unseen parts, suggesting that their memory representation is not sufficiently detailed to rule them out as distractors.

## INTRODUCTION

Every day, we are surrounded by a rich and complex visual world. Previous research has shown that we are able to memorize a large amount of visual information (*Standing, 1973*). In recent years, these findings have been replicated for objects (*Brady et al., 2008*) and scenes (*Vogt & Magnussen, 2007*; *Konkle, Brady & Alvarez, 2010*), indicating a massive memory capacity for complex stimuli. These experiments typically use a similar procedure, in which a large number of stimuli are presented in the study part and then identical views (or foils) are later queried in the testing part.

Although a high visual capacity was observed in many studies, one might naturally ask how detailed the memories about each item are. This level of detail is usually termed

Corresponding author
Filip Děchtěrenko,
dechterenko@praha.psu.cas.cz

fidelity. In a memory study with thousands of objects (*Brady et al., 2008*), participants correctly recognized target objects when presented with similar foils, which differed only in their state. Moreover, in the free recall paradigm using drawings, people show remarkable memory of objects and their spatial location (*Bainbridge, Hall & Baker, 2019*).

This high-fidelity representation is challenged by findings from *Vogt & Magnussen (2007)*. In their study, they presented participants with 400 pictures of doors, and they observed a high recognition accuracy of approximately 85%. In a follow-up manipulation, they removed all of the surrounding details of the doors, and the recognition accuracy decreased by 20%.

The observed high recognition performance might overestimate the true capacity due to the type of experimental probing. In general, participants are able to more easily provide a correct response in a two-alternative forced choice (2AFC) task than in a Yes/No task. Studies directly comparing both methods (*Cunningham, Yassa & Egeth, 2015*; *Andermane & Bowers, 2015*) reported comparable results. However, *Cunningham, Yassa & Egeth (2015)* concluded that the high performance in the original study by *Brady et al. (2008)* might also be explained by gist-based representations, as the same set of subjects showed significantly lower recognition performance in the Yes/No task than in the 2AFC task. Thus, the representation of the studied item might be noisy, and the performance in the memory studies might result from gist-based representations rather than storing detailed information about the stimuli. A similar finding of a rather low-level detailed representation was observed in the incidental encoding paradigm (*Draschkow et al., 2019*), in which participants' memory was tested using the Yes/No task. Again, the accuracy was lower than that in the original study by *Brady et al. (2008)*. The current literature thus provides conflicting evidence regarding what people actually remember when memorizing a larger set of stimuli.

One way to approach the question of what we remember is to use more ecologically valid stimuli. The typical stimuli used in visual memory experiments are individual objects (*e.g.*, *Brady et al., 2008*) or photographs of scenes (*e.g.*, *Konkle, Brady & Alvarez, 2010*). However, in our daily experience, we are surrounded by a continuous stream of visual information. We can remember objects or specific views, but the explicit boundaries given by a photograph's frame are missing. Herein, we distinguish between *scenes* (real-world combinations of objects) and *photographs* (providing views of scenes). We are interested in overlapping photographs that share some common content. The usage of overlapping photographs allows us to study repeated exposure and assess how individual photographs are integrated into coherent memory. Although people are able to recognize previously presented viewpoints (*Hock & Schmelzkopf, 1980*), they do not integrate the viewpoint with information about viewed objects within the scenes (*Varakin & Loschky, 2010*). During the memorization of overlapping photographs, participants usually look more at the previously viewed parts rather than exploring novel, previously unseen parts (*Valuch, Becker & Ansorge, 2013*). We can view the distinction between scenes and photographs as a part of a hierarchy and take it one step further by asking about smaller parts of the photographs (here called patches) and how well they are remembered when the photographs are repeated.

When memorizing a photograph, people usually have trouble remembering the view boundary. The problem with remembering view boundaries is prominent in boundary extension (BE) studies (*Intraub & Richardson, 1989*), where people incorrectly claim they remember seeing objects beyond the border of the photograph. According to the source-monitoring account on BE (*Intraub, 2012*), people automatically perform additional processes when seeing a photograph–they amodally complete missing parts of objects or surfaces, categorize the scene and activate contextual associations. The observed error of commission is a result of confusing actual memories with these additional inferences. Due to the design differences, the actual BE effect is not likely relevant for our observations. However, we share a common idea that people try to position the observed content (patches in this case) into a larger context (representation of the photograph). Similarly, people try to position the observed photographs into larger scenes they imagine (as in BE). Making assumptions about the scene beyond the view is not limited to BE paradigm. In particular, people can make judgments about more distant views, *e.g.*, whether two views are from the same panoramic view (*Robertson et al., 2016*), but their performance is limited unless extra cues are provided and both views overlap. Thus, participants are sensitive to scene layout to a limited extent (as shown by *Sanocki & Epstein, 1997*, and others).

In the present study, we studied visual memory of scenes from different perspectives. As previous studies showed conflicting evidence regarding what people actually remember of each studied photograph, we worked with larger photographs and presented participants with only partial views of these photographs. Thus, we subsequently presented an alternative view partially overlapping with the original view and tested whether this second presentation provided a memory advantage for later recognition tests. The unseen parts of the photograph were also a suitable source of distractors, as they showed the same semantic category and usually similar texture (in the case of natural scenes) or scene-related objects (of bar stools in restaurants). This method provides a stricter measure of human recognition memory and allowed us to select foils similar to the presented stimuli, which might be difficult in the case of whole scenes. Because this approach may increase the number of false alarm errors, we decided to use the Yes/No paradigm in all experiments.

In Experiment 1, we studied whether participants were able to recognize parts of the presented photographs. We were interested in whether the repeated presentation of the whole photograph would lead to increased recognition of smaller parts, similar to the findings in the study by *Konkle, Brady & Alvarez (2010)*. Additionally, querying participants with lures selected from either unseen parts or presented parts allowed us to assess the level of detail of recognition performance ("This area looks like part of the kitchen I saw" *vs.* "This area is the particular part of the kitchen I saw").

Our paradigm allowed us to better determine whether the performance in recognition tasks is more a result of vague resemblance of the memorized photograph or more a result of the detailed memory of a particular photograph. The main purpose of Experiment 1 was to explore the general performance in recognizing stimuli by their smaller parts using estimates from signal detection theory framework. This general paradigm was further

extended in two follow-up experiments. In Experiment 2, we focused on whether the recognition accuracy is explained by the properties of individual images, namely, their memorability. We estimated the average recognition accuracy (memorability) for each image using a deep neural network, and we tested to what extent participants recognized parts of highly memorable images or were lured by similar foils. Finally, in Experiment 3, we performed eye-tracking analysis and assessed how the repeated content was studied. We tested whether participants preferred to rescan the area they had viewed during the first presentation or explore the new areas and how the dwell times on parts of an image correlated with the actual recognition accuracy.

## EXPERIMENT 1

Previous research has shown that people are able to recognize previously presented photographs with high accuracy (*e.g.*, *Standing, 1973*; *Konkle, Brady & Alvarez, 2010*). However, to correctly recognize a photograph or reject previously unseen photographs, the content of the photograph needs to stand out from the remaining presented stimuli either semantically or perceptually (*Lukavský & Děchtěrenko, 2017*; *Standing, 1973*; *Konkle, Brady & Alvarez, 2010*; *Bylinskii et al., 2015*). Can we recognize a presented photograph based on only a part of the image? One would naturally suspect that memory accuracy would be lower for smaller parts, as memory for whole photographs can benefit from memorizing multiple parts. However, even when recognizing smaller parts of presented photographs, people may still have only vague feelings of familiarity. Therefore, are we able to correctly reject parts of photographs that have not been previously presented? If so, this may suggest that the memorized photographs are stored in great detail, while in the case of high false alarms, the stored representation may actually have minimal details.

The main goal of Experiment 1 was to assess the human memory of parts of photographs, which we approached in two ways. First, we tested whether people could learn from a larger photograph and respond about its parts. We presented participants with photographs, but in the recognition phase, we used small patches from the presented photographs instead of asking about the whole presented photographs. Participants were tested after every five trials (immediate recognition) and after presentation of the complete set of photographs (delayed recognition). Second, we tested whether people were able to learn more about the smaller parts when they encountered the larger photograph twice, either in the identical form or in the overlapping views. In the overlapping views, only half of the scene content was presented in the second viewing, and new content was exposed.

This first experiment was designed to assess general performance on this paradigm and later elaborated in Experiments 2 and 3. All experiments were approved by the Ethics Committee of the Institute of Psychology, Czech Academy of Sciences (approval number PSU-203/Brno/2020).

## Materials and Methods

### Participants

Twenty-eight subjects (mean age = 21.54 years, SD = 2.35 years; 7 males) participated in the experiment. All participants had normal or corrected-to-normal vision, and none had

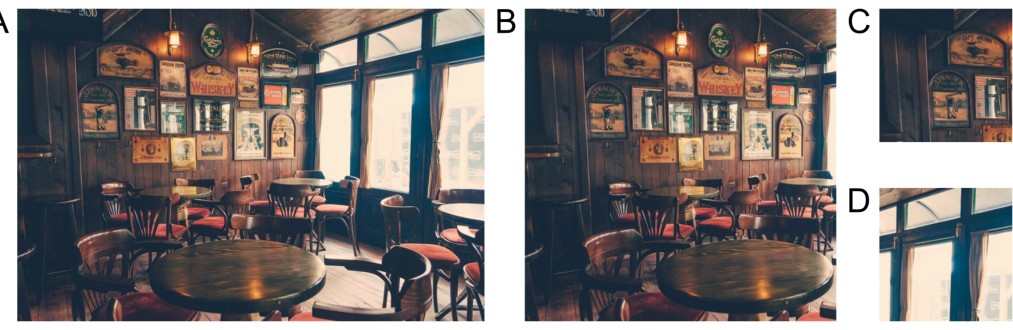

**Figure 1 Example of visual images.** (A) Source image used to create the visual stimuli, (B) The view presented to the participants. In the recognition phase, we asked either about patches of the original view (C) or about patches beyond the original view (D). The images used in this figure for peer review were replaced with similar illustratory photographs for publication due to copyright concerns.

previously participated in this type of experiment. The subjects provided written consent before the start of the experiment.

### Stimuli and apparatus

For the experiment, we obtained pairs of photographs that overlapped in 50% of their content. Briefly, we selected a set of photographs and split each of them into two overlapping views (the scheme is visualized in Fig. 1). The photographs were obtained from the SUN database (*Xiao et al., 2010*). We selected 11 categories to contain both natural and man-made scenes (indoor: candy store, bar, shopping mall, and atrium; outdoor: basilica, cabin, and slum; natural: creek, forest, canyon, and mountain). As the SUN database contains many photographs with various formats, we selected only landscape-oriented photographs larger than 900 × 600 px. This resulted in a dataset with 1,183 photographs.

These photographs were resized to 900 × 600 px while maintaining the original aspect ratio. If one dimension was larger than the specified size, we cropped the remaining area and visually inspected the photograph for possible cropping artifacts. The resulting photograph was divided into thirds of 300 × 600 px each, and the training stimuli consisted of two adjacent thirds (600 × 600 px). For the creation of test stimuli, we extracted smaller patches from each photograph. The patches were 256 × 256 px (43% × 43% of the stimulus or 18% of the area) located either in the upper or lower part of the photograph (randomly selected). From each photograph of 900 × 600 px, we created all six smaller patches (upper/lower × left/central/right patch). Each participant was presented with a random set of both training stimuli (on average 6 photographs per category, SD = 3.78) and corresponding test patches.

The experiment was programmed in PsychoPy (*Peirce, 2007*) and presented on a 22″ LCD screen (1920 × 1080 px). The participant sat approximately 50 cm from the screen, and the head position was not controlled.
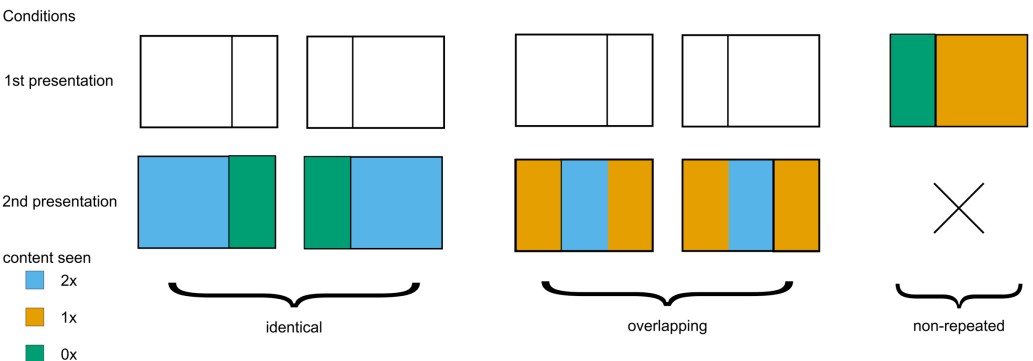

**Figure 2** **Three types of experimental stimuli.** Repeated and overlapping stimuli were presented twice, while non-repeated trials were presented only once. Color denotes how many times a part of an image was presented.

## Design

The experiment consisted of two parts: the study part and the testing part. In the study part, participants were presented with the photographs and tested for immediate recognition. In the testing part, participants were tested for the recognition of photographs from the whole study part. We manipulated two factors in this study: level of repetition of the photograph during the study part and the selection of the smaller patch with respect to the corresponding learned photograph.

Regarding the level of repetition, the presented photographs were classified into one of three types (see Fig. 2). First, *identical trials* (2 × 55 images) featured the identical photograph presented twice per participant. Second, *overlapping trials* (2 × 55 images) presented two different views from the same original photograph that shared the middle portion (left view followed by right view or *vice versa*). Third, *non-repeated trials* (110 images) contained unique photographs with no repetition.

Smaller patches were selected either from the presented parts of photographs, unseen parts of presented photographs or from completely novel photographs. Different patch conditions were present in immediate and delayed recognition.

For immediate recognition, the testing trials featured 66 patches of three conditions (22 trials per condition). The patch was either from the presented part of the non-repeated photograph (*old*), from an unpresented part of the non-repeated photograph (*lure*), or from a novel, unseen photograph (*new*). Each of the three patch types was presented 22 times.

For delayed recognition, the testing trials featured 200 patches of four possible conditions (50 trials per condition). Similar to immediate recognition, patches from parts of images that participants saw once (*old-1*), patches from the unseen parts of the presented stimuli (*lure*) and patches from novel, unseen images (*new*) were presented. We also presented patches from the parts of images that participants saw twice (*old-2*). The number of patches per condition and their relationship to the presentation trial conditions are summarized in Table 1. For all patches, we evenly distributed whether the patch was selected from the upper or lower part of the original image.

**Table 1 Breakdown of presentation trials for patch selection.**

| Patch type | Presented images | | | | | | New images |
|---|---|---|---|---|---|---|---|
| | Identical trial | | Overlapping trial | | Non-repeated trial | | |
| | Presented part | Unseen part | Central part | Noncentral part | Presented part | Unseen part | Unseen part |
| Old-2 | 25 | | 25 | | | | |
| Old-1 | | | | 25 | 25 | | |
| Lure | | 25 | | | | 25 | |
| New | | | | | | | 50 |

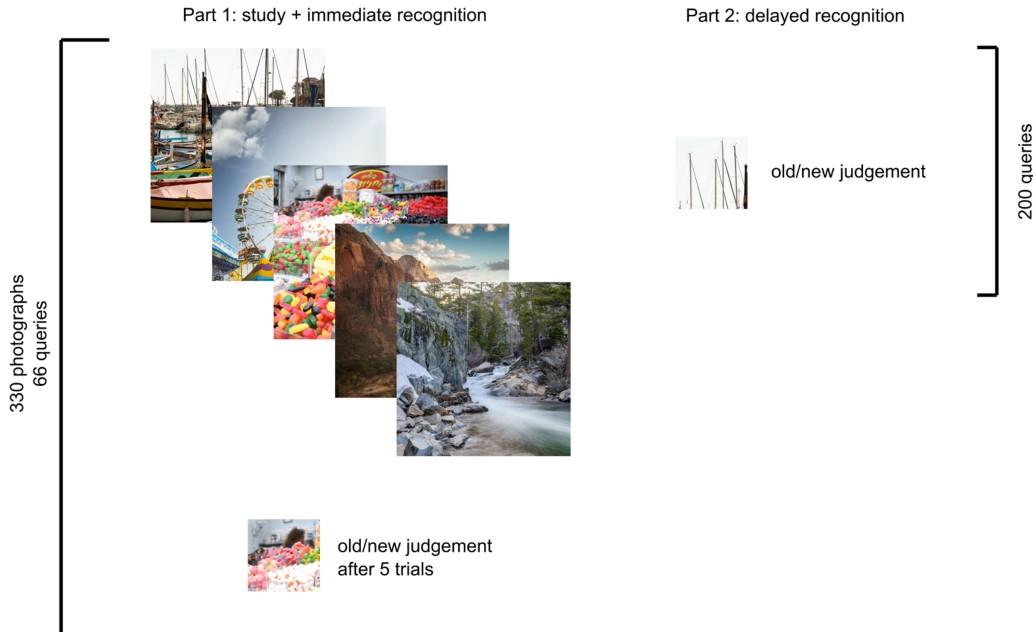

**Figure 3 Experimental scheme depicting examples of stimuli in each part.** In the study part, patches were selected from the previous five images, while in the testing part, the queried patches were selected from the whole set. The images used in this figure for peer review were replaced with similar illustratory photographs for publication due to copyright concerns.

## Procedure

As shown in Fig. 3, the study part consisted of 396 trials: 330 presentation trials in which a photograph (600 × 600 px) was shown in the center of the screen for 3 s for participants to study and 66 testing trials. One testing trial was presented after each of the 5 presentation trials, and the task was to decide whether the presented patch was part of one of the preceding five images (with unlimited time). Participants were instructed to memorize the presented photographs as accurately as possible, and they were informed that they would be presented with smaller patches and that their task would be to respond, regardless of whether they saw the exact patch within the photograph. Participants were informed that some patches may look similar to previously presented photographs but that they should really respond only if that patch was exactly the one that was

presented previously (we did not tell them that some patches would be from the unseen parts of the photographs). Participants were also aware that in the study part, patches would be related only to the 5 previously seen photographs, while they would later be queried for the whole set. Additionally, they were informed that we were interested in recognition accuracy, not in the speed of the response (although they were asked not to spend too much time on each trial if they were unsure).

The order of trials was randomized for each participant with the constraint that each block of 5 trials contained at least one non-repeated trial.

In each testing trial, we showed a single patch, and participants were asked whether they had seen this exact patch during the study part. The patch was shown in the center of the screen, not at its original position during the presentation. Participants responded with either the left key (patch was shown) or right key (patch was not shown), while verbal cues (words *old* and *new*) were provided below each image to reduce the possibility of response mistakes.

Although we asked participants to prioritize accuracy over response time, we analyzed the response time to detect the potential speed *vs.* accuracy trade-off.

### Data analysis

All materials and scripts are available at the Open Science Framework (https://osf.io/469er/). Data were analyzed with the R software (*R Core Team, 2019*) using generalized linear mixed models with a binomial link function (lme4 package, *Bates et al., 2015*), *i.e.*, hierarchical logistic regression. We tested the accuracy of different patch types separately in immediate and delayed recognition. Both situations were tested with a model containing two random factors (varying intercepts for both participants and images), one fixed factor as the independent variable (patch type) and the participant's response (correct yes/no) as the dependent variable. For immediate recognition, the patch type had three levels (new, lure, and old), while four levels were used for delayed recognition (new, lure, old-1, and old-2).

Models were compared with a null model containing only random factors using the likelihood ratio test. To further compare the relative difference in false alarms between the lure and the new patches, we reran the regressions on the subset of data containing the new and lure patches only and expressed the relative difference between conditions using the odds ratio. Similar approach was performed in case of delayed recognition to express the difference in odds ratios between old-2 and old-1 patches. The differences from the chance level were tested using one-sample *t*-tests. Finally, for delayed recognition, we tested the differences between presentation trial types using a two-sample *t*-test for each of the patch types (*e.g.*, for old-2 patches, we tested the differences between repeated and overlapping trials).

We also analyzed the data with the signal detection theory (SDT) approach to assess the changes in sensitivity between immediate and delayed recognition (*Green & Swets, 1966*). We used the additivity property of the sensitivity measure (d') to measure the perceptual distance between stimuli separately for immediate and delayed recognition. We first

computed z-scores for responses of type "seen" using the approach described by *Macmillan & Creelman (2004)* to compute d'. The distances between the z-scores defined the sensitivity, and the bias was set to zero. We computed d' and bias on the responses for novel patches (corresponding to false alarms) and old patches (corresponding to hits) to determine the differences in sensitivity and bias between the immediate test and delayed test. We corrected the hit rate and false alarm rate by adding 0.5 to all cells to avoid perfect performance (which we observed in three participants) as suggested by *Macmillan & Creelman (2004)*. Changes in d'/bias were measured using a paired *t*-test. The expected decrease in sensitivity between immediate and delayed recognition corresponded to decreased memory capabilities over time. On the other hand, theoretical changes in bias would represent changes in response pattern; in particular, lower bias values correspond to more false alarm responses.

## Results

For immediate recognition, participants performed well in identifying new patches they had not seen previously (accuracy 81% or 19% false alarm rate). The accuracy for old patches (73%) was lower. Participants struggled when responding to the lure patches from areas they had not seen (accuracy 41% or 59% false alarm rate). The model treating patches differently had a significantly better fit than the null model with random factors only ($\chi^2(2) = 247.37$; $p < 0.001$). Furthermore, participants scored significantly greater than chance (chance level: 50%, $p < 0.001$) under all conditions. When comparing the conditions directly, participants were 6.6 times more likely (95% CI [5.1–8.7]) to score a false alarm for lure patches than for new patches.

For delayed recognition, the accuracies were lower: 69% for old-2 patches, followed by new (59%) and old-1 (58%). The FA rate for lure patches was 49%. The differences between patch types were significant ($\chi^2(3) = 91.09$; $p < 0.001$). Participants scored significantly greater than chance for new, old-1, and old-2 patches ($p \leq 0.003$), but the performance was not significantly different from random guesses for lure patches ($p = 0.639$). Pairwise comparisons revealed that participants were 1.4 times more likely to record a false alarm (95% CI [1.2–1.7]) for lure patches than for new patches. Similarly, participants were 1.6 times more likely to correctly recognize the target when the patch was presented twice (95% CI [1.4–1.9]). Because each of the test patches was selected from two types of trials during the presentation part in the delayed recognition (see Table 1), we assessed differences in recognition accuracy and the FA rate under these conditions. None of the differences were significant ($ps \geq 0.293$), indicating no effect on accuracy (*e.g.*, when old-2 patches were selected from the identical *vs.* the central part of overlapping trials) or false alarms (when lures were selected from unseen parts of identical trials *vs.* non-repeated trials). The results of both parts are depicted in Fig. 4.

Based on the SDT framework, the perceptual distance in immediate recognition between old and new patches was 1.52, that for old and lure patches was 0.40 and that for lure and new patches was 1.12. The distance between lure and old patches is also
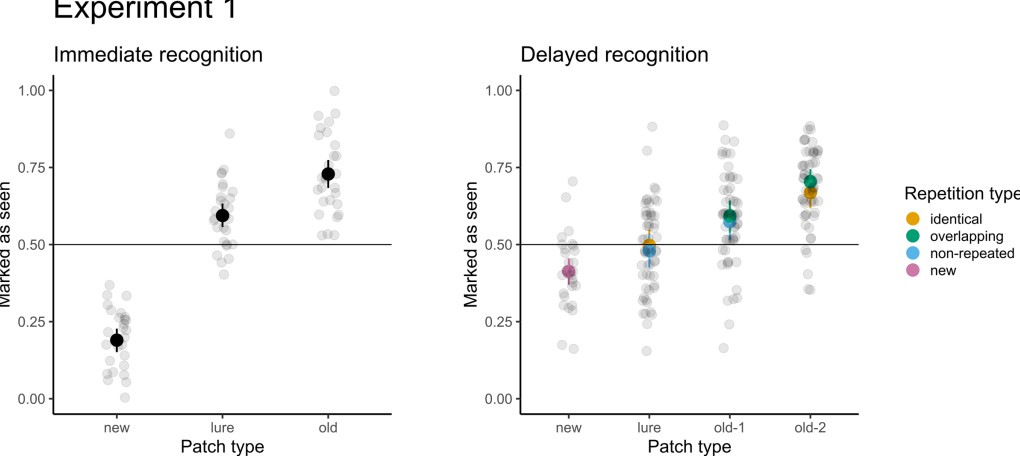

**Figure 4 Accuracy in immediate (left) and delayed (right) recognition.** Vertical lines denote bootstrapped standard errors of the mean. Individual points denote the average score per participant.

known as pseudo d' (*Dosher, 1984*). For delayed recognition, the perceptual distances were smaller (old-2–old-1: d' = 0.28, old-1–lure: d' = 0.25, lure–new: d' = 0.20).

The differences in performance calculated as d' were significantly decreased between immediate recognition (mean d' = 1.55, SD = 0.6) and delayed recognition (mean d' = 0.44, SD = 0.19, $t(27) = 10.51$, $p < 0.001$). Regarding the response pattern, the bias was not different between immediate (mean bias = 0.14, SD = 0.33) and delayed recognition (mean bias = 0.01, SD = 0.33, $t(27) = 1.58$, $p = 0.125$). Thus, participants did not systematically prefer any of the responses (old/new).

The correlation between accuracy and reaction time was not significant for immediate recognition and delayed recognition ($p \geq 0.446$).

## Discussion

Two main findings emerge from the results. First, although participants exhibited high recognition accuracy for presented patches and correctly rejected novel patches, they also recorded a high false alarm rate when presented with a patch from the hidden part of a presented photograph. This finding shows a remarkable level of uncertainty about these distractors. Second, the recognition accuracy was higher for patches that were shown twice, but the accuracy was not significantly different when the same photograph was seen twice or when two photographs with overlapping content were viewed. Similarly, we observed a similar false alarm rate for lures irrespective of whether they were selected from the unseen parts of identical or non-repeated trials.

We obtained similar results by analyzing the percentage of correct responses and using the SDT approach. As expected, participants' sensitivity decreased after the delay. Interestingly, we did not observe significant differences in bias after the delay. The results show that participants' implicit preference for either old or new responses was balanced and did not change over time.

Děchtěrenko and Lukavský (2022), *PeerJ*, DOI 10.7717/peerj.13187                **10/23**

## EXPERIMENT 2

Experiment 1 showed that although participants were able to recognize small areas from the presented stimuli, they were also prone to false alarms when they were queried with unseen parts of photographs. This suggests that participants appeared to base their decision only on rough semantics and perceptual similarity to the presented photographs. In research concerning visual memory, it is generally observed that some photographs are more recognizable than others. This recognition score is commonly referred to as memorability (*Isola et al., 2011*). Memorable photographs have higher perceptual organization (*Goetschalckx et al., 2019*) and cannot be predicted by simple visual features such as color (*Isola et al., 2011*) or spatial frequency (*Bainbridge, Dilks & Oliva, 2017*). An unresolved question is to what extent memorable photographs consist of memorable parts and regions (*Khosla et al., 2012*). In the context of our experimental design, are memorable photographs more recognizable by their parts and correctly rejected in the case of unseen parts? To answer these questions, in Experiment 2, we studied how the capability to recognize the whole scene corresponds to the recognition of individual patches. We utilized the same protocol (and the same photographs) for all participants, which allowed us to compute the average recognition score for each presented patch (memorability score) and average false alarm rate for unseen parts (lureability score). The idea of using the lureability score was derived from previous findings that people show consistency in false alarms as well (*Bainbridge & Rissman, 2018*).

Although the optimal approach would be to obtain memorability and "lureability" scores for whole photographs and compare them with patch data, this experiment would require a separate large memory study, as our selected dataset did not contain behavioral memorability scores. Instead, we estimated the memorability of the whole photographs with a convolutional neural network (*Khosla et al., 2015*).

We proposed several hypotheses about the relationship between the memorability of whole photographs and their parts. According to previous research, memorable photographs (*Isola et al., 2011*) should be easier to remember because their content differs from that of images from similar categories. Therefore, we expected that the hit rate and correct rejections should be high for old and new patches of highly memorable images (*Bylinskii et al., 2015*). For the lure patches, two competing hypotheses exist. First, highly memorable images are memorable because of their content; therefore, the false alarm rate (incorrectly marking images as old) should be low (accuracy should be high). Alternatively, highly memorable images are memorable because of their style or gist. Consequently, the accuracy should be low because they would be prone to false alarms caused by distractors from unseen parts, which extends the presented content.

### Materials and Methods

#### *Participants*

Twenty-eight subjects (mean age = 21.39 years, SD = 3.46 years; 4 males) participated in Experiment 2. All participants had normal or corrected-to-normal vision, and none had participated in Experiment 1 or a similar experiment previously. Again, all subjects provided written consent before the start of the experiment.

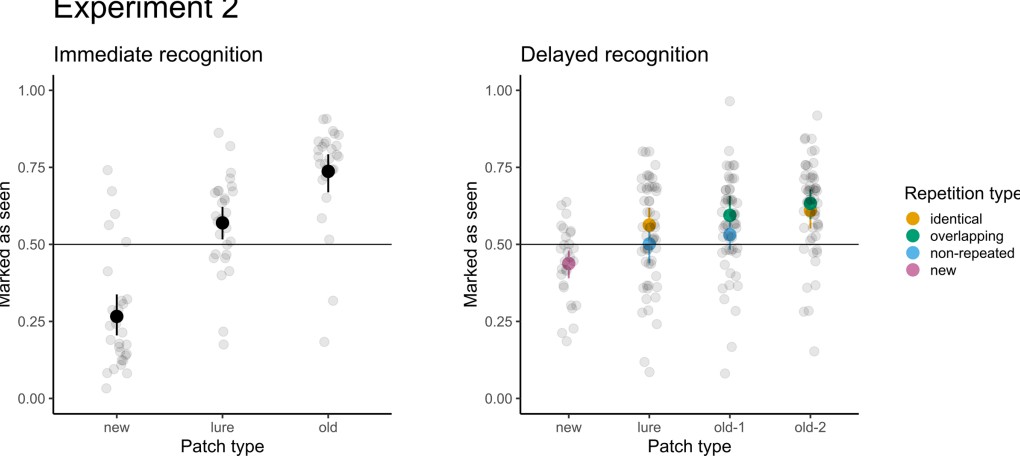

**Figure 5 Accuracy for immediate recognition and delayed recognition in Experiment 2.** Vertical lines denote bootstrapped standard errors of the mean. Individual points denote the average score per participant.

## Stimuli and apparatus

The apparatus was identical to that used in Experiment 1. The stimuli were similar to those used in Experiment 1: the only difference was that all participants were presented with an identical set of photographs.

## Procedure and data analysis

The procedure was identical to that in Experiment 1. We computed Fleiss' κ to measure the agreement of participants' responses. Memorability was estimated using the pretrained MemNet net (*Khosla et al., 2015*). We estimated memorability for both the left and right two-thirds of the original photographs, and because the difference in estimated memorability between both parts was small (mean difference = 0, SD = 0.04), we used the average value of those two scores as the memorability score for the whole photograph. For each patch, we computed the average hit rate (memorability, for old and new patches) or false alarm rate (lureability, for lure patches). We correlated the model-estimated memorability for whole photographs using MemNet with the average patch memorability/lureability.

## Results

The results for both immediate and delayed recognition were similar to those obtained in Experiment 1 (Fig. 5). The participants showed significant agreement in accuracy (immediate recognition: Fleiss' κ = 0.19, z = 30.1, $p < 0.001$; delayed recognition: Fleiss' κ = 0.14, z = 38.1, $p < 0.001$). We observed variability in the average recognition/false alarm rate between images (see Fig. 6)[1].

The model-estimated memorability ranged from 0.48 to 0.83 (mean = 0.66, SD = 0.07). When averaged per each category, highest memorability was observed for candy store (mean = 0.79, SD = 0.05), while mountains showed lowest memorability (mean = 0.60,

[1] We do not show similar plots for test patches as the use of four patch types made the plot difficult to read.

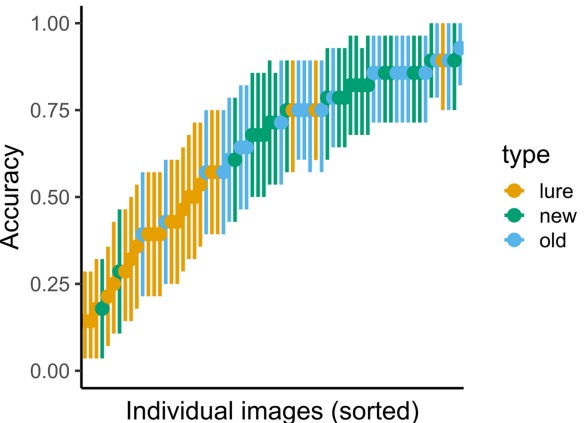

**Figure 6 Variance in accuracy for immediate recognition in Experiment 2.** Vertical lines denote bootstrapped standard errors of the mean.               

SD = 0.06). Averaged model-estimated memorability values per category were similar to the behavioral memorability values described in *Bylinskii et al. (2015)*.

For immediate recognition, we observed a positive (though nonsignificant) relationship between the model-estimated memorability of the photograph and correct recognition of the patch ($r = 0.40$, $p = 0.064$) and a negligible correlation between memorability and correct rejections in the case of new distractors ($r = 0.06$, $p = 0.285$) when aggregating responses per patch. In the case of lureability (average false alarm rate), the correlation with model-estimated memorability was nonsignificant for all patch types ($|rs| \leq 0.2$; $ps \geq 0.164$). Taken together, model-estimated memorability for whole scenes did not explain the recognition performance for small patches[2].

---

[2] Similar estimates were observed for the Spearman correlation coefficient.

## Discussion

Experiment 2 confirmed the results of Experiment 1. A visual inspection of the patches that showed high consistency in memory performance revealed that participants probably recognized the photographs using either highly memorable objects in the photograph or the semantic uniqueness of the photograph ("there was only one white mountain"). We estimated the memorability using a pretrained neural network and assessed whether this model-estimated memorability of the whole photograph affected the memory performance for parts (patches). In our experiment, the model-estimated memorability of the whole photograph was not a good predictor of remembering its part (all correlations were small, with the exception of immediate recognition). The lureability or tendency for false alarm errors was not explained by the model-estimated memorability of the whole photograph. Consequently, we were not able to distinguish between our competing hypotheses about memorable photographs. Our main finding is that the model-estimated memorability of the whole photographs does not correspond to the recognition of the respective parts. Our main limitation is the usage of model-estimated memorability instead of using empirical estimates. The model was trained to estimate memorability as the hit rate, and it did not naturally capture the average similarity in false alarm rates.

# EXPERIMENT 3

The previous two experiments consistently showed that people have difficulty with correctly recognizing photographs by their smaller parts. We found that repeated presentations of the stimuli lead to increased recognition accuracy, but we found no benefit of showing overlapping content instead of showing whole photographs twice. Thus, we aimed to investigate how people study the repeated content and measured their eye movements. The natural question that we aimed to address was whether participants utilize this experimental manipulation in their strategy (and observed recognition accuracy). Do they spend time fixating on the shared content (potentially merging the representations together) or do they tend to devote more time exploring the novel content? This strategy may also be reflected in their recognition performance, as *Olejarczyk, Luke & Henderson (2014)* observed that participants showed higher recognition accuracy during incidental encoding of the parts that they fixated on longer. Similarly, *Valuch, Becker & Ansorge (2013)* found that people fixate more often on the parts they have fixated on in previous presentations.

## Materials and Methods

### Participants

Twenty-one subjects (all female, mean age = 21.86 years, SD = 3.05 years) participated in this experiment. None had participated in the previous experiments, and all participants had normal or corrected-to-normal vision.

### Stimuli and apparatus

Experiment 3 was programmed using custom scripts in MATLAB with the Psychtoolbox extension (*Brainard & Vision 1997*; *Cornelissen, Peters & Palmer, 2002*; *Kleiner, Brainard & Pelli, 2007*; *Pelli, 1997*). Trials were presented on a 22″ LCD display with a resolution of $1680 \times 1050$ and a 60 Hz refresh rate. Eye movements were measured using Eyelink II (SR Research, Ottawa, Canada). Participants were seated 50 cm from the monitor, and the head was positioned on a chin rest. We used the same stimuli and protocols as in Experiment 1 (each participant was presented with random selection of stimuli). In this setting, the images subtended $19.4° \times 19.4°$, and the test patches subtended $8.3 \times 8.3°$.

### Procedure and data analysis

The eye tracker was calibrated using 9-point calibration before each part of the experiment, and drift correction was performed before each trial. Otherwise, the procedure was identical to that used in Experiments 1 and 2. All stimuli (both whole photographs and smaller patches) were presented in the center of the screen.

Fixation and saccades were detected using the algorithm supplied with the eye tracker. Our main goal was to analyze whether participants fixated more on the overlapping parts of the photographs than the nonoverlapping parts. Additionally, we report simple descriptive statistics of eye movements (fixation duration and saccade amplitude) for overall picture.

### Simple descriptive statistics

Our main dependent variables were the fixation duration and saccade amplitude. We retained all tracking data, irrespective of whether participants fixated on the presented stimuli[3]. We analyzed eye-tracking data from the presentation trials (identical, overlapping, and unique). We did not analyze the eye-tracking data in the test trials due to differences in the presentation times.

We used two linear mixed models with either saccade amplitude and fixation duration as dependent variables and the type of trial (identical, overlapping, or unique) and first/second presentation as fixed factors to determine the significance of differences in the fixation duration and saccade amplitudes between trial types and repeated presentations. Participants and stimuli were treated as random factors, which allowed us to fit varying intercepts for each participant and photograph. We compared the significance of differences in both parameters using a model comparison approach (as in Experiment 1), but the continuous nature of the dependent variable (and thus using a linear mixed model, instead of a generalized model) allowed us to report the results using F tests with Satterthwaite's method (*Giesbrecht & Burns, 1985*; *Kuznetsova, Brockhoff & Christensen, 2017*), which is more accessible to the reader given the similarity with nonhierarchical statistical methods. We obtained similar results using both methods.

### Overlapping content

In this analysis, our main dependent variable was dwell time. We did not define areas of interest based on the locations of the patches for two reasons. First, the patch areas were small; thus, the results would be more prone to spatial measurement error during eye tracking. Second, the patches were selected from fixed coordinates within the photograph without any notion of the semantic content. Thus, binning of the fixation data into these particular areas would be arbitrary, and conclusions would be difficult to infer. Consequently, we computed the dwell times separately for each third of the image only.

We analyzed the dwell times with respect to the first and second presentations to assess whether participants fixated more on portions of the images that were already presented or on novel content. For clarity, the meanings of novel and overlapping content reversed the eye-tracking data for the stimuli displaying the right two-thirds of original image as they were measured in the left-two thirds[4]. Using this reversal, we treated the data as we always showed first for the left two-thirds of the original image, while in the second presentation of overlapping trials, we are showing the right two-thirds. We denote the thirds of the original image as *lateral* (left third), *central*, and *contralateral* (right third) to make our results more salient. All identical and unique trials and the first presentation of overlapping trials thus displayed the lateral and central thirds, while the second presentation of overlapping trials displayed the central and contralateral thirds.

When participants viewed the photographs for the first time (irrespective of whether the stimuli were identical, overlapping, or unique), we expected null differences in dwell times between halves (central third *vs.* lateral third). When viewing photographs for the second time, participants might employ different viewing strategies in the case of identical and overlapping trials. When viewing identical photographs for the second time,

[3] We repeated the analysis for the data with the fixation outside the presented stimuli discarded, but the results were similar.

[4] Without this approach, we would need to report the analysis separately for trials showing the left two-thirds and trials showing right two-thirds during the first presentation. We obtained similar results from this separate analysis.

**Table 2 Fixation durations and saccade amplitudes for each trial type and presentation.** Means with SD shown in parentheses.

| Variable | Trial type | 1st presentation | 2nd presentation |
|---|---|---|---|
| Fixation duration | Non-repeated | 276.51 (42.10) | – |
| | Identical | 270.92 (36.14) | 280.69 (48.60) |
| | Overlapping | 274.50 (40.41) | 279.40 (48.94) |
| Saccade amplitude | Non-repeated | 4.10 (0.71) | – |
| | Identical | 4.11 (0.70) | 4.07 (0.80) |
| | Overlapping | 4.14 (0.70) | 4.06 (0.80) |

do participants use the same strategy (*e.g.*, because one half contains more informative content) or do the dwell times change (*e.g.*, because they want to explore the parts they fixated on less during the first presentation)? For overlapping trials, do participants fixate the central third more, as it shares content with the previous presentation, or rather do they spent more time on the novel (contralateral) third? We used linear mixed models with two fixed factors denoting whether fixation was occurring in the central or lateral/contralateral part and whether it was the first or second presentation to test these hypotheses. We used dwell time as our outcome variable and participants as a random factor. We used all data from all three types of trials (as they always showed lateral and central thirds) to test the hypothesis regarding the first presentation, while we separately analyzed data for identical (as lateral and central thirds were presented in both presentations) and overlapping trials (as the second presentation showed central and contralateral thirds) to test the hypotheses regarding the dwell time pattern in the second presentation.

Finally, we also correlated the averaged patch recognition accuracy with the dwell time in image third from which the patch was selected. Because Experiments 1 and 2 showed low memory accuracy in delayed recognition, we used the recognition accuracy from the immediate recognition condition only.

## Results

### Accuracy

The memory accuracy in immediate recognition showed a similar trend as that observed in Experiments 1 and 2 (immediate recognition: lure: 41% or 59% FA, old: 78%, new: 83%).

### Fixation duration and saccade amplitude

The basic descriptive statistics of eye movements are presented in Table 2. Fixation durations were not different between the types of trials ($F(2, 66,047) = 2.88$, $p = 0.056$), while for identical and overlapping trials, fixation durations were significantly longer in second presentations than in the first presentations ($F(1, 68,218) = 13.15$, $p < 0.001$, $\Delta = 6$ ms, 95% CI [3–10]). Fixation durations were similar in all types of trials for the first presentation (identical, overlapping, and non-repeated). The interaction between type of trial and first/second presentation was not significant ($F(1, 68,209) = 1.83$, $p = 0.177$).

For the saccade amplitudes, we did not observe differences between types of trials ($F(1,$ $60,162) = 0.057$, $p = 0.945$), presentation ($F(1, 62,376) = 1.43$, $p = 0.232$), or the interaction ($F(1, 62,366) = 0.335$, $p = 0.563$).

### Overlapping content

An examination of the eye-tracking data revealed significant differences in fixation locations. When viewing the photographs for the first time, we observed a strong preference for the central third (mean = 1,551 ms, SD = 603 ms) over the lateral third (mean = 1,204 ms, SD = 594 ms; $F(1, 8,967) = 762.66$, $p < 0.001$). This pattern was repeated during the second presentation for identical trials (central: mean = 1,568 ms, SD = 653 ms; lateral: mean = 1,196 ms, SD = 612 ms; $F(1, 2,180.1) = 191.61$, $p < 0.001$). In the case of overlapping trials, we again observed a preference for the central third (mean = 1,543 ms, SD = 601 ms) over the novel contralateral part (mean = 1,195 ms, SD = 606 ms; $F(1, 2,215.1) = 188.85$, $p < 0.001$). This preference for the central part was not observed for recognition accuracy, as we did not detect a correlation between immediate recognition accuracy and dwell time ($r = -0.01$, $p = 0.801$). Taken together, these results indicate a strong bias for the central parts (without a behavioral effect on recognition). This finding was surprising, as the participants were fixating on the central part of the original image, even during the first presentation of the image.

We speculated that the preference for the middle part of the whole image might result from the artificial crop introduced by creating the stimuli. Photographers usually do not take photographs randomly but rather position interesting content in the central part of a photograph. The artificial crop could disrupt the natural boundaries of the photographs, and this artificial border might be distracting for the participants. We conducted a small control experiment in which participants ($N = 8$) were presented with the stimuli from Experiment 1 and their task was to respond whether the photograph originally continued to the left or to the right to assess this hypothesis. Each participant was presented with 593 stimuli and correctly detected the crop in the image in 72% (SD = 5%) of cases (50% guessing level). Therefore, people are able to estimate the composition of a particular view relative to a larger unseen photograph.

## Discussion

The eye-tracking analysis of the overlapping paradigm showed that fixations tend to be slightly longer when observing identical photograph a second time. The main finding from the eye-tracking experiment was preference for the central part of the original photograph. Participants were able to detect the central part, even in the first presentation, without knowing in which direction the scene should continue. The stimuli were presented in the center, and this preference for the central part meant that participants were looking slightly to the left or to the right from the center of the screen (depending on whether the original stimuli continued to the left or right). By conducting a small control experiment, we also found that the preference for the central part might result from the artificial crop introduced when creating the stimuli. Photographers do not take photos at random but position the interesting content in the center. This finding emphasizes the

importance of precise stimuli selection. Small artificial modifications, such as cropping an image, can influence eye movements.

## GENERAL DISCUSSION

In a series of three experiments, we explored to what extent people make correct recognition statements about partial scene views. Our experiments differed from other visual memory experiments in three aspects. First, we did not show identical views in the recognition phase (both immediate and delayed). We showed partial views (referred to as patches) covering 18% of the original photograph. Second, we used the unpresented parts of the photographs as a source of foils that generally showed similar semantic content. Finally, we repeatedly presented the same photographs (either identical or overlapping views) and tested whether they would be recognized better.

Our experiments show that people recognize photographs from their partial views well if they are queried immediately (within a minute), and the recognition decreases to nearly chance level when the participants are queried after the end of the study phase. Although people recognize photographs from their partial views, the performance is worse than in previous studies, where participants were queried with stimuli identical to those presented in the training phase (*Konkle, Brady & Alvarez, 2010*; *Vogt & Magnussen, 2007*; *Andermane & Bowers, 2015*; *Lukavský & Děchtěrenko, 2017*). Smaller patches contain less information than the whole photographs, which may be one source of errors. Patches were not taken to highlight a specific object, which might make recognition more difficult, especially after longer delays. Numerous studies have shown a high capacity of visual memory for objects (*Brady et al., 2008*) and scenes (*Konkle, Brady & Alvarez, 2010*), including the capacity to correctly recognize state changes in objects. However, matching the stored image representation with a presented partial view was difficult. Our experiments show that some search is possible within short-term memory but substantially impaired when long-term memory is tested.

Memory performance was significantly affected when people were asked about partial views from unseen parts of the photograph. People were particularly prone to false alarm errors (59% in Experiment 1). This finding further supports our argument that the stored representation is either not sufficiently detailed or is difficult to search. People might instead rely on high-level information (semantic information about the scene) or heuristics about the photograph's texture and style.

The high proportion of false alarms in lure patches is consistent with the boundary extension effect and the source-monitoring account (*Intraub, 2012*). We provided people with foils from unseen parts of the photograph, which potentially share many properties with the original view. Moreover, people might be able to detect a common layout of the underlying scene. *Gottesman (2011)* showed that partial views of scenes prime distance judgments in areas that were not visible in the original view. However, our setup substantially differed from the usual BE experiments. Namely, we presented people only with the patch from the unseen part during the recognition phase, while BE is usually observed when presenting the wide-angle view of the original photograph (showing both the previously seen and the unseen parts). Our lure patches depicted content 3.5% to 46.5%

beyond the border, while the differences studied in BE experiments are more subtle. The BE account would likely predict more false alarms in repeated presentations, which we did not observe. Our experiment does not exclude the potential role of BE. In accordance with the BE account, our results further emphasize that our memory is not sufficiently detailed to exclude more difficult lures, and we may be confused by a similar style or underlying layout.

In addition to using smaller patches instead of whole scenes, our design also employed the Yes/No paradigm instead of 2AFC, as in previous studies. Using the Yes/No paradigm makes the task more difficult, as shown both theoretically (*Macmillan & Creelman, 2004*) and empirically (*Cunningham, Yassa & Egeth, 2015*; *Andermane & Bowers, 2015*). A reasonable assumption is that the use of 2AFC in our task would lead to higher accuracy.

We expected that repeated presentations would provide additional viewing time and lead to better encoding. This expectation was confirmed, and the additional presentation increased the accuracy from 58% to 69% (in Experiment 1). Additionally, we were interested in whether the additional presentation would exert a different effect if identical views or overlapping views were presented. We did not propose a specific hypothesis; the overlapping views might exert multiple effects. If people fuse their representations into a single form, this process may cost additional time and effort, which are not spent on encoding details, resulting in poorer memory than identical repeated views. Alternatively, fusion may lead to deeper processing of the input and better encoding along levels of the processing framework. However, we consistently observed no difference. Our experiments may not have been sufficiently sensitive to measure the difference or people did not fuse the photographs into a single representation.

We also presented the same set of stimuli to all subjects, which allowed us to compute patch memorability (for seen patches) and patch lureability (for unseen patches). In the case of both immediate and delayed recognition, the correlation between patch memorability (or lureability for unseen patches) and memorability estimated from a pretrained neural network was small. Therefore, either the model-estimated memorability of the whole photograph was not a good estimate for the memorability of its regions or memorability (estimated as the hit rate) does not guarantee rich memory representation. In other words, people do not notice changes even in highly memorable photographs. This claim is supported by recent findings from *Broers & Busch (2019)*, who explored recollection and familiarity with respect to image memorability. They reported large differences in stimuli–some highly memorable stimuli showed high recollection, while the remaining stimuli showed a similar relationship for familiarity. A visual inspection of patches with high false alarm rates and high hit rates also supports this claim. Patches with a high hit rate either contained a highly informative object, or the whole photograph was highly distinct from the remaining stimuli. Patches with high false alarm rates either contained textures similar to seen parts or contained objects that were also present in the presented part. The results of visual inspection are subjective in nature and need to be verified in further studies. Taken together with the finding that model-estimated memorability was similar for scenes sharing only 50% of the content, the

memorability score estimated by the neural network (and showing correlation with large-scale memorability studies) did not appear to correspond with the amount of encoded details.

The eye-tracking analysis showed slightly longer fixations in second presentations. Otherwise, the participants' eye movement patterns were similar, regardless of whether the stimuli were presented twice or whether they overlapped. The most surprising finding was that participants fixated more on the part corresponding to the central part of the original photograph, even when the photograph was presented for the first time. This finding dominated all other possible viewing strategies, such as fixating on the overlapping parts more or fixating on the novel content during the overlapping condition during the second presentation. We confirmed this preference in a small control experiment, which indicated that participants were able to detect the artificial crops introduced when creating the stimuli. These results confirm that participants are aware of the overall scene layout and could use this information for further judgments (*Gottesman, 2011*). Furthermore, the findings emphasize the importance of careful stimuli selection and controlling for possible confounders. Photographs of visual scenes are usually taken with a certain spatial layout in mind, which would easily be disrupted when a photograph is cropped. We did not detect increased recognition of the parts with higher dwell times. This result contradicts findings from *Olejarczyk, Luke & Henderson (2014)*, who reported higher recognition accuracy for parts on which participants fixated more. In our case, we did not measure dwell times for small patches but rather for larger semantic areas (whole thirds), which might mask the possible benefit of a prolonged viewing time. The observed strong preference for the central third might hide more subtle effects.

## CONCLUSIONS

The capacity and fidelity of visual memory have been important research topics in recent years, and the fidelity or nature of stored information is an unresolved question.
To address the importance of storing details or simple gist-based representation, we used a novel methodology with photograph patches and repeated presentations. Taken together, the study has two main findings.

First, people can recognize the remembered photographs from their partial views. However, their memory performance is worse than usually observed, and they are highly prone to false alarm errors in distractors, which were selected from the unseen parts of the presented photographs. We suggest that this vulnerability to false alarms arises from the semantic or gist information people use in their memory decisions. The probability that the photograph will be recognized from a partial view cannot be explained by model-estimated memorability for the whole photograph. The eye-tracking analysis confirmed that the probability that a photograph will be recognized by its partial view does not depend on the time spent looking at the stimulus.

Second, participants showed higher accuracy for parts of scenes that they saw twice. A surprising finding was the existence of a small fixation bias toward the center of the original, larger scene from which the particular view was selected. This result suggests that participants are sensitive to the general layout of the scene.

### Funding

This research was supported by Czech Science Foundation grants (GA 16-07983S and GA 20-06894S) and RVO68081740. The funders had no role in study design, data collection and analysis, decision to publish, or preparation of the manuscript.

### Grant Disclosures

The following grant information was disclosed by the authors:
Czech Science Foundation: GA 16-07983S, GA 20-06894S and RVO68081740.

### Competing Interests

The authors declare that they have no competing interests.

### Author Contributions

- Filip Děchtěrenko conceived and designed the experiments, performed the experiments, analyzed the data, prepared figures and/or tables, authored or reviewed drafts of the paper, and approved the final draft.
- Jiří Lukavský conceived and designed the experiments, authored or reviewed drafts of the paper, and approved the final draft.

### Human Ethics

The following information was supplied relating to ethical approvals (*i.e.*, approving body and any reference numbers):

All experiments were approved by Ethics committee of Institute of Psychology, Czech Academy of Sciences (approval number PSU-203/Brno/2020).

### Data Availability

Data and scripts are available at Open Science Framework (OSF): Dechterenko, Filip, and Jiri Lukavsky. 2021. "False Memories When Viewing Overlapping Scenes." OSF. Dataset. https://osf.io/469er/.

Stimuli for the study are available at SUN: SUN Database: Large-scale Scene Recognition from Abbey to Zoo". J. Xiao, J. Hays, K. Ehinger, A. Oliva, and A. Torralba. IEEE Conference on Computer Vision and Pattern Recognition, 2010.

The SUN Database is continuously updated. To view a static copy of the database, see the benchmark sections: https://groups.csail.mit.edu/vision/SUN/hierarchy.html.

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
