# Peer review of "False memories when viewing overlapping scenes"

_PeerJ, doi:10.7717/peerj.13187_

## Round 0.1 · original submission · Major Revisions

Your submission has been reviewed by two experts who have provided extensive, detailed, critical constructive comments on all parts of the manuscript. Both reviewers opined that the research you report is interesting and deserving publication, but that the current version of the manuscript includes a large number of issues/shortcomings that require careful, thoughtful, and fairly extensive reflection and revision. I agree with this appraisal.

The reviewers’ comments provide clear guidance on the revisions that are required to make this manuscript publishable. Instead of summarizing their comments, I only will provide a brief overview here and add my own few additional comments.

One of the reviewers noted that “the clarity of the manuscript needs to be improved throughout, as the paper is somewhat difficult to read in its current form”. The reviewer goes on to suggest that the MS needs “a more targeted introduction to focus more on the points that are directly relevant to the present study —that is, [the manuscript needs an introduction focused on] “what readers need to know to understand why the study was done, what questions it [was] aimed to answer, and what work others have done that are relevant for this question and methodology”. To get you started, the reviewer has offered specific suggestions on how you might reframe the Introduction.

The reviewers also identified another aspect related to the clarity of the manuscript: The inconsistent use of different words/terms used for the same concept (e.g., “study part” and “study phase”; “cut” and “crop"; scene “parts” versus “patches”, etc.). I agree with this observation: The use of inconsistent labels introduces confusion and makes the manuscript unnecessarily difficult to read/understand.

The Method also needs to be clearer and more complete. One reviewer noted that the manuscript does not specify if the “same scenes were used for each of the conditions. Figure 1 seems to suggest it, but it isn't actually stated. If so, [was] the scene-condition pairing counterbalance[d] across participants? If not, was there any attempt to equate the scenes in the different conditions?”

I had similar questions about the Method, about the absence of information that would be required for an exact replication and for a fully informed interpretation of the findings. Questions: How many pictures were selected for the study, what criteria were used for selecting pictures, were all pictures segmented in the exact same manner, were the patches used for 0 vs 1 vs 2 presentations counterbalanced, was the use of the distracter-patches counterbalanced, etc. And as asked by the reviewers, if the scene-condition pairings were not counterbalanced across participants, “was there any attempt to equate the scenes in the different conditions”?

The reviewers also expressed serious concerns about the result sections, “in particular, [about] the lack of statistics to support conclusions that are drawn”. Statistical evidence in required to support several claims that are made in the manuscript, And as noted by the reviewers, the manuscript would be more effective if it were re-framed “to emphasize more of what you did find rather than what you didn't, particularly with respect to your results for lures”.

Reviewer 1 ·

Basic reporting

Overall, this paper addresses an interesting question using a clever design. In particular, I commend the authors on designing this paradigm using unseen parts of old scenes, as I think this will be a useful paradigm for the field. Although I know that interest is not relevant for meeting the journal criteria, I do want to add that both the finding of high false alarms to unseen regions, and the finding that central regions of the original scenes were preferentially fixated even when they were not the center of the presented images, are very interesting and answer important questions.

My biggest concern is that the clarity of the manuscript needs to be improved throughout, as the paper is somewhat difficult to read in its current form.

First, different terms are used in different places to refer to the same concept (e.g., “study part” and “study phase”; “cut” and “crop"; scene “parts” versus “patches”), which makes it harder to extract the main points.

There are also more subtle instances, for example “We found that fixations were biased towards the center of the original larger photograph and that this bias was repeated during the second presentation in both identical and overlapping views.” The first presentation is never mentioned, so the reader has to infer from the end of the sentence that the first half of the sentence was referring to the first presentation. Solving these types of issues would result in a paper that is substantially easier to read.

Furthermore, the manuscript would benefit from careful editing for English usage, in both syntax and nonstandard word choices. For example of the latter, “panoramatic” is used rather than the term used by the panoramic study being discussed, “stimuli” is used as singular and plural in different places, and “cut” is used instead of “cropped” in most places.


My impression is that the authors provided generally sufficient background information. My only specific suggestion with respect to citations is to discuss Valuch, Becker, & Ansorge (2013), as their paper uses a very similar method.

The composition of the introduction and discussion could be improved. In particular, I felt that the transitions between points and paragraphs in the introduction often lacked logical flow, and it wasn’t clear to me why some points were being discussed, or what questions the authors were hoping to answer with their study. I would suggest writing a more targeted introduction to focus more on the points that are directly relevant to the present study—that is, what readers need to know to understand why the study was done, what questions it is aimed to answer, and what work others have done that are relevant for this question and methodology.

In the beginning of the discussion, it was not clear why Andermane and Bowers (2015) was discussed in such detail, particularly with respect to their delays. Was the point to show that other studies have found that memory decreases after a delay?


I found the figures to be excellent. My only suggestion for the figures is to make the points larger in figures 4 and 5, as they are very small currently.

Experimental design

The analytic approach seemed to be excellent based on the description in the method, but I feel it was not consistently well executed or described in the results. For the most part, the design was well thought out, with my only major hesitation being the memorability estimation (see section 3).

Furthermore, as in the rest of the manuscript, I strongly recommend the authors improve the clarity of writing in the method and results section.

Some of the areas that lack clarity do so in a way that hinders the ability to understand the method and analyses, for example:
1. “The model contained two random factors (intercepts for participants and images), one fixed factor (patch type) and the response (correct yes/no) as dependent variables.” It is unclear in this case which variable is the outcome measure.
2. “Vigilance patches” is used only once in the manuscript (in the results), and it is not clear what it means.

I think that using “central,” “lateral” and “contra-lateral” in place of “identical, overlapping, and novel” introduces unnecessary confusion that makes things harder for the reader. I realize this was likely done because the significant result pertained to fixations being central, but this can be stated in a more efficient way without the need for new terms.

Lastly, please clarify the Experiment 3 statement that “We did not limit dwell times to the patch areas, because their content could be arbitrary and people might better investigate their surrounding context.”

Validity of the findings

My most serious concerns pertain to the results sections; in particular, the lack of statistics to support conclusions that are drawn. I recommend that statistics be provided, or the conclusions be removed. Some instances:
1. The main conclusion from Experiment 3 (e.g., “these results indicate a strong bias for the central patches“ and “The main finding from the eye-tracking experiment was preference for the central part of the original image”) does not appear to be supported by any statistical tests. It seems that the authors only provide means and standard deviations for this data. This is particularly concerning given that the discussion states that “the results are near the threshold level for significance”
2. The authors used generalized linear mixed effects models with model comparison, but in some places, only presented the results of the comparisons, without providing the significance or direction of the independent variables (e.g., in the Fixation duration and saccade amplitude section of Exp 3, it is stated that adding trial type improved the model fit, but it is not stated which direction these effects occur or what conclusion we can draw from them).
3. It is stated in the Exp 3 discussion that “the average duration of the second fixation was slightly longer”, however, this is not mentioned in the results or supported by statistics.
4. In the general discussion, line 551, it is stated that “Patches with high false alarm rates either contained textures similar to seen parts or contained objects that were also present in the presented part”, however, this does not seem to be supported by any data.
5. In the final conclusions section, it is stated that “they are highly prone to false alarm errors in distractors closely matching the style of the original stimulus,” however, it does not seem like that claim is supported, as distractors were looked at as a single entity, not analyzed by style match.

I would avoid discussing the directions of the nonsignificant effects at such length. For example, in Experiment 2 lines 324-331, only the first two nonsignificant correlations are explicitly referred to as such, whereas the rest of the nonsignificant correlations are referred to as if they were significant effects (“correlation was slightly negative”). I am concerned that this may mislead readers who are skimming and not reading the statistics, or more importantly, the general public who may read this paper without a knowledge of statistics.


It seems like there is an implicit assumption in the paper that neural nets are a good proxy for memorability, but they are a rough estimate of memorability at best (i.e., a correlation of .64). In particular, it is not known how well those neural networks estimate memorability in the present subset of stimuli. Because memorability was not directly assessed in this study, I highly recommend that the authors greatly temper their conclusions with respect to the null memorability correlations, and change all uses of “memorability” to “model-estimated memorability.” This issue is particularly concerning in the abstract, where it is not mentioned that the memorability measure is not actually memorability.

·

Basic reporting

The article presents important empirical data in clear and professional language. However, there is a significant alternative possible interpretation of a major findings of the study. The "false memory" for lures is not presented well in my opinion. The discussion of the Boundary Extension work is very relevant here and should be included in the Introduction as well as research on scene layout. You may want to consider Gottesman (2011), which I think is particularly relevant here, but in any case research on scene layout should be discussed in the introduction.
(Gottesman, C. V. (2011). Mental layout extrapolations prime spatial processing of scenes. Journal of Experimental Psychology: Human Perception and Performance, 37(2), 382–395. https://doi.org/10.1037/a0021434)

There are a few stylistic issues (as well as some lack of clarity in the design and procedure that will be addressed later).
Stylistic issue:
The discussion of Experiments 2 and 3 at the end of the introduction doesn't flow well and interrupt the flow of the paper. I recommend shortening it and making it more relevant given the overall introduction.
Minor points:
-Lines 182-186 are not presenting information in consistent way if it is "old 2- 25 patches from identical trials and 25 from..." should it not be "old 1- 25 patches from... and 25 patches from... and the same for "lure..."?
-Figure 4 's heading doesn't state "Experiment 1"

Experimental design

The overall theoretical frame work and design are good with some significant problems.
Most important, as mentioned above, there is a significant alternative theoretical framework that is mostly ignored in the article yet provides a very important alternate explanations for the results. The authors use "lures" which are patches or parts of the scenes that were not presented to the participant with the expressed purpose of having distractors that are supposedly matched for "content and style". However, there is a strong alternative view to this argument that should be presented upfront. These patches are not just similar images (indeed there is no clear evidence that they are in some way matched for content and style). They are actually parts of the actual scene from which the photographs were taken. Extensive Boundary Extension literature (which is only briefly mentioned in the discussion) suggest that people understand photographs of scenes (as well as drawings, and real physical partial views through apertures) as parts of bigger scenes. These patches are not just similar in gist or style, they are parts of the same scenes presented. In addition to the same textures and gist they also show parts of the layout of the views presented. The importance of layout, in particular, is ignored for the most part in the article and yet it is arguably a significant part of scene representation. Gottesman (2011) for example, showed that partial views of scenes primed distance judgments in areas of the scene that were not visible in the prime view. The results obtained in this study are strongly consistent with this view, particularly the strong FA rate for lures. There are other weaker indications like the tendency to focus on the central part of the scene (in Exp3), as if people can predict where the center is and therefore arguably, have some idea (though obviously not a perfect one) of what the "unavailable" parts show.
Thus, the FA for the lure can arguable be evidence for much more than just memory for gist and superficial characteristics, indeed there is no strong indication that it is memory for gist rather than layout or other characteristic specific to that particular scene (though not the view presented originally) .

In this vein, it would be also interesting to see if the FA for lures is stronger in repeated v. non repeated trials (that data should be available).

Design clarification needed:
-It is not clear if the same scenes were used for each of the conditions. Figure 1 seems to suggest it but it isn't actually stated that I could find. If so, is the scene-condition pairing counterbalances across participants? If not, was there any attempt to equate the scenes in the different conditions?
-Was the selection of lower or upper parts of the scene distributed similarly across lure and old conditions? (see line155) This may be important in how much of the scene layout can be understood from the patch.
- The method section doesn't report recording RT (there is mention of it in the results only)
- Line 188 reports that the patches were always presented in the center. I could find a similar statement for the full photographs. I assume it was but, particularly given the discussion in Exp. 3, I recommend stating outright that the full photographs were also presented in the center, if that was so.
-Figure 2 seems to show a patch where the objects are bigger than in the full photograph, if this is incorrect, please fix the figure, if not this may be a problem, please explain in the text.
- The conceptual significance of bias in the SDT analysis should be explained more. Was a difference expected? what would it mean?
- Line 311- Introduces the terms "vigilance patches" v. "test patches" without defining them. Is there need for new terminology here? if so define.

Validity of the findings

The results are well analyzed overall. with a few issues that I think should be addressed.
- First, the conclusion that "participants base their decisions only on rough semantic and perceptual similarities" (line 270) is arguable because the lure patches are not just similar, they are parts of the same scenes as the presented images. There is no direct test of similarity that is reported.
-The discussion of correlations (lines 326-332) should be more carefully worded as none of the effects are statistically significant.

Smaller issues:
-line 204- not clear what mean is referred to- how was it calculated? (right section memorability - left. for each picture, averaged across pictures?)
-I would like to see descriptions of the trends in the figures in the text itself (missing for several of the figures. e.g. line 314 refers just to "differences"; what differences were observed?)
-Figure 7 is particularly confusing, I was not clear what you were trying to demonstrate in it, and it wasn't very effective.

Additional comments

I think this research is very interesting and it would be more effective if you re-frame it to emphasize more of what you did find rather then what you didn't, particularly with respect to your results for lures. I am torn between classifying this as minor or major revisions. I think the procedure and analysis were probably fine even in the areas that were not clear in the manuscript but just in case my inferences were incorrect I would like to see it spelled out.

---

## Round 0.2 · Major Revisions

I carefully read the revised manuscript and the detailed notes from the reviewer. The reviewer and I are in agreement in our appreciation of the extensive revisions already made to the manuscript. and also in agreement that there still is room for improvement. The reviewer recommends further minor revisions. My recommendation is the same. However, I underscore that ‘minor’ does not mean the same as ‘unimportant’.

A number of improvements are required still to make this manuscript fit for publication in PeerJ. First, and as emphasized by the reviewer, the manuscript still needs further label sorting. The manuscript includes too many labels for items that were displayed to participants – photographs, images, patches, scenes, etc. -- and their inconsistent use makes reading very difficult if not impossible. The reviewer provides detailed constructive guidance on this issue. My add-on recommendation: you might use one label, like scene, when talking about what we see in the course of a day, and use the word photograph only when talking about the stimuli or stimulus-parts used in the experiments. And get rid of synonyms like image, and don’t use stimulus as a synonym for photograph or image or scene.

The other parts of the manuscript in need of further clarity are the questions asked by each experiment. I infer from the manuscript that Exp 1 was about recognition memory for photograph patches (RMPP). But the MS says nothing about specific question pertaining to RMPP. Was it (the question) as simple as: we wanted to know about the limits of RMPP, in d’ & beta? Was the plan to compare RMPP to recognition memory for other parts, like episodic memory for parts of a melody, or episodic memory for parts of a sentence or story or list? To make each experiment more interesting, please identify the precise question or questions that generated it, and do so in reference to related research.

Also required for getting this MS above the publication threshold is a more informative and complete method section for each experiment. The method sections are ok as far as they go, but they are not complete because they fail to give info about critical independent variables (IV) that were manipulated. As a reader, I have to infer IVs from the manner in which they were operationalized. This will not do. Without IVs, it is difficult to compare the current research with other related research, and difficult to create guiding research hypotheses.

This manuscript is the most basic unadorned laboratory research report I have seen in a long time. I am familiar with this type of writing when the purpose is publication in an archive. However, for the purpose of reaching PeerJ readers, I believe the research reported in this manuscript needs to be contextualized more clearly in contemporary research. The manuscript is a fair distance from reaching this goal; it needs more connections with recent research, with references to this research.

I read this manuscript as being about – from the 10 mile up perspective -- episodic memory for everyday visual encounters with the world? In the laboratory version of this domain, Experiment 1 is about the capacity and fidelity of human recollection of certain types of photos. The manuscript should answer to: Why look into this? No-one ever investigated? No-one ever investigated with photographs like yours? What theories drive this type of applied memory research? The reader needs context for motivation to read this new MS, as well as for interpreting the new findings.

I was unable to access figures, other than Figure 1. Not concerned because I saw them in the last review cycle.

A final major comment: The manuscript needs to give more information about instructions given to participants. Memory performance is determined by sensory-perceptual, cognitive, attentive and automatic processes, and at least some of these are influenced by participants’ mental set, with depends to some extent on tasks and task instructions. This understanding is recognized in the MS, by its description of the instructions given to participants about how to do their task (speed and accuracy are important). But the MS has nothing to say about the specific instructions for each task. What were participants told about parts of photos, relation between the different portions used for hits and false alarms, etc.?

A few minor comments:
Ll225: Please provide a reference, for questions about this method of aggregating data?

Did you collect any response time data in any experiments?

L~450 I appreciate the terms lateral, central and contra-lateral.

L490+ I am surprised by the dwell time data. You report lots of null effects in the manuscript. For context, please describe what kind of effect you would have needed for detecting it with adequate power.

·

Basic reporting

I think the manuscript is greatly improved. Additional research and possible theoretical viewpoints are presented and the terminology used is much clearer. For the most part, the terminology is more sensitive to the distinction between discussing truly different scenes (e.g., Bar v. mountain stream) and discussing unseen parts of previously viewed scene (e.g., the right part of the bar compared to the left part of the same bar). The use of “Image, versus image part” works well here but there are a couple of places where the use of the term scene is still confusing.
This is a difficult issue to define: What is a “scene”? What do you mean by it? Is the scene just what was shown in the picture or is it the environment that was photographed? I can take two pictures of my kitchen showing different parts of it; are they two pictures of the same scene (possibility 1)? If not, are they different scenes in the same way as one of these pictures would be different compared to a picture taken in my friends’ kitchen (possibility 2), even if we had very similar furniture and fixtures? I want to clarify that this is where I think the significance of the BE work really lies. It is not in the actually boundary memory error, which as you say is not necessarily relevant here., but in the possibility that BE is evidence for the viewer’s greater understanding of the actual scene, beyond what is presented (as in possibility 1). The viewer obviously knows they didn’t see the whole scene but when trying to place the boundaries accurately (as in the BE paradigms) or when trying to judge if they’ve seen a patch of a scene (in your paradigm), both different but somewhat challenging tasks, they are revealing their access to a greater representation of the scene, than was visible in the view presented in the photograph. I believe that is the relevance to your findings.
Whether you agree with this argument or not, it would be less confusing if you didn’t use “scene” as synonymous with “the photograph of the scene”, this happened rarely in this version but it still does. Here are a couple of placed where I found this:
74: “The usage of overlapping stimuli allows us to study repeated exposure and
measure how individual scenes are integrated into coherent memory.”
The words “Individual scenes” suggest to me different scenes, like the kitchen and the hallway. Consistent with the rest of your discussion, I think you are really saying something like
“individual scene parts”

110: “We were interested in whether the repeated presentation of the whole scene
111 would lead to increased recognition of smaller parts similarly due to repeated exposure of the
112 whole stimuli (Konkle et al., 2010).”
First, as discussed above if you are taking the “possibility 1” definition of scene, you are not presenting the whole scene but the “the whole photograph” or “the whole stimulus”. Photographs rarely, if ever, show a whole scene.
Moreover, can this sentence be broken up? Do you even need the second part after “due”? It seems repetitive to me.

This is my main concern for clarity. Beyond this, there are several small clarity issues, particularly in the introduction. I will discuss them in Additional Comments as I believe they are mainly typographical in nature, though I may be wrong in places where I was confused.

Experimental design

The experimental design description was greatly clarified compared to version 1 and reveal solid experimental design. Difficulties and imperfections were addressed adequately.

Validity of the findings

Many complex cumbersome sections and sentences have been un-packaged compared to Version 1, and valuable clarifications were added. For the most part, the results are interesting and alternative possible explanations are offered.

There is just one point that I found difficult to fit with the rest of the work. I am not sure if it’s an error of expression or is attempting to make a point that is not coming across. In Experiment 2 you said:

302 Experiment 1 showed that participants were unable to easily recognize small areas from the
303 presented stimuli.

I think this is a pretty unconvincing statement. I guess this comes down to what how you define "easy" I would argue that 73% accuracy for old patches is not negligible. You contradict this statement in line 548 yourself. If you want to make this argument here you should qualify it better.

Additional comments

I think what I have below are all minor issues but they need to be addressed to make the manuscript clearer. I am listing the parts I stumbled on; I do not want to rewrite your manuscript but I put some suggestions of my own, just to illustrate what I feel the difficulty is. Of course, I may have missed the point in some of these places, as I was confused by them, so my suggestions may not work, but I hope they will help you to understand my confusion.

Abstract
25 were observed. In Experiment 2 (N=28), neither the recognition accuracy for an image part nor
26 the tendency for false alarms correlated with the memorability of the corresponding larger image
27 estimated by pretrained deep neural network. Finally, in Experiment 3 (N=21), we repeated the

I recommend breaking up this sentence. Something like “In Experiment 2 (N=28), the memorability of the larger image was estimated by a pretrained deep neural network. Neither the recognition accuracy for an image part nor the tendency for false alarms correlated with the memorability”’

Introduction:
40 : a large number of stimuli are presented

“A large number is presented”, or “large numbers are presented”

116 Experiment 2, we focused on the consistency of correctly responding to whether the recognition
117 accuracy is explained by the properties of individual images, namely, their memorability.

This sentence is not clear. Is the “consistency of correctly responding to” not the same as recognition accuracy? Is it not enough to just say “we focused on whether the recognition
accuracy is explained by the properties of individual images, namely, their memorability.” What am I missing?

197 shown), while verbal cues were provided below each image to reduce the effect of response
198 mistakes.

“Verbal cues” can mean different things. What does it here? If it’s something like “press left for ‘patch was shown’ press right for ‘patch was not shown’” I recommend putting it under that patch in Figure 2 as it would have appeared on the participant’s screen, or clarify in a different way.

226 patches (or in hits between old-1 and old-2)

Why is this statement in parentheses? Is seems very important in itself.

241 to avoid perfect performance (which we observed in three participants)
Last part of this sentence is stated twice.

242 Macmillan & Creelman (2004). Changes in d’/bias were measured using a paired t-test. The
243 theoretical meaning of the expected decrease in sensitivity between immediate and delayed
244 recognition corresponds to decreased memory capabilities over time. On the other hand,
245 theoretical changes in bias would represent changes in response pattern; in particular, lower
246 bias values correspond to more false alarm responses.

I asked you for the theoretical meaning here, so I like this clarification but you don’t need the actual words “theoretical meaning”. They make the sentences cumbersome. “the expected decrease in sensitivity between immediate and delayed recognition would corresponds to decreased memory capabilities over time…. etc,” will work better.

Experiment 2

312 color (Isola, et al., 2011) or spatial frequency (Bainbridge, Dilks, & Oliva, 2017). Additionally,
313 memorability has been defined not only as correct recognition but also as a false alarm rate

What false alarm rate? low? high?

Experiment 3

456 To make our results more salient, we will denote the
457 thirds of the original image as lateral (left third), central, and contralateral (right third) to make
458 our results more salient.

Repeated phrase.


Figure 1 caption:
one too many “about” in “we asked about either about patches…”

---

## Round 0.3 · Minor Revisions

First, let me apologize for the delay in issuing a decision; the prior editor dropped out of the process, creating some delay. However, I have now carefully reviewed the manuscript and your responses (in all their iterations), and have also received a re-review from Reviewer 2, which is positive.

I believe that you have successfully addressed the issues raised by the prior editor and the reviewer on the last version of the manuscript, and the effort has resulted in a manuscript that I find clear and coherent. I particularly appreciate the consistent use of the terms photograph vs. patch.

I would like to raise three minor points. I realize this is late in the game for a new editor to be requiring substantial changes, so I will cast those as “for your consideration” rather than as required.

1. Per prior editor’s comment E7: Rather than describing what instructions subjects received, can you simply provide the instructions that they saw (or that were read to them)? That would be clearer, and would also promote reproducibility of your methods.

2. Similarly, Line 257 states that all materials are available on OSF. However, I saw figures, code and datasets, but not the stimulus (photos and patches) that were used. Can these be made available for reproducibility?

3. Lastly, I was surprised that Experiment 2 did not provide any descriptives related to the model results, for example estimated memorability. I realize a “memorability estimate” may be in arbitrary values, so it may not be so meaningful to report, e.g., an average memorability across photos, but is there any useful information such as variance/range that could be provided? If variance/range were low, that could partly explain the lack of correlations observed.

Accordingly, I am issuing a decision of MINOR REVISION. Please respond to the three points above, which you may either adopt, or provide a short response explaining why you cannot (or choose not to) comply.

This is also your opportunity to correct the few remining errors (typos) identified by Reviewer 2.

When you resubmit, it is likely (although cannot be guaranteed) that I will be able to quickly issue a final decision of ACCEPT without requiring further re-review.

Again, apologies for the delays, and I hope you will agree that the manuscript has been made much stronger and more impactful by the reviewers’ input and by your revisions.

·

Basic reporting

The manuscript successfully addressed the concerns raised by prior reviews. The language and terminology are clear. The impetus for the study is convincing and enough context is provided.

Experimental design

The design and procedure have be successfully clarified.
typographical error: There is an extra word "about" in the heading of Figure 1
"In the recognition phase, we asked [about] either about..." remove []

Validity of the findings

The results are interesting and compelling. There are several significant effects and the nulls effects are also interesting in themselves. The power needed for the null results to achieve significance is not reported but the authors make a compelling argument that the effects are so small as to make achieving significance very unlikely.
The conclusions are warranted from the results.
Typographical error: Line 578 the word "this" repeated twice at the end.

---

## Round 0.4 · accepted · Accept

I am pleased to be able to accept the manuscript for publication. I realize that this has been a long process, but I think (and I hope you agree) that the manuscript has been greatly improved by your implementation of the reviewers' suggestions, and I think it will make a nice contribution to the literature.